# The Valorisation of Olive Mill Wastewater from Slovenian Istria by Fe_3_O_4_ Particles to Recover Polyphenolic Compounds for the Chemical Specialties Sector

**DOI:** 10.3390/molecules26226946

**Published:** 2021-11-17

**Authors:** Kelly Peeters, Ana Miklavčič Višnjevec, Esakkiammal Sudha Esakkimuthu, Matthew Schwarzkopf, Črtomir Tavzes

**Affiliations:** 1InnoRenew CoE, Livade 6, 6310 Izola, Slovenia; sudha.esakkimuthu@innorenew.eu (E.S.E.); matthew.schwarzkopf@innorenew.eu (M.S.); crtomir.tavzes@innorenew.eu (Č.T.); 2Andrej Marušič Institute, University of Primorska, Muzejski trg 2, 6000 Koper, Slovenia; 3Faculty of Mathematics, University of Primorska, Natural Sciences and Information Technologies, Glagoljaška 8, 6000 Koper, Slovenia; ana.miklavcic@famnit.upr.si

**Keywords:** polyphenolic compounds, olive mill wastewater, extraction techniques, Fe_3_O_4_ particles, magnetic collection, adsorption and desorption, quantitative and qualitative analysis

## Abstract

Olive oil production using three-phase decanter systems creates olive oil and two by-products: olive mill wastewater (OMWW) and pomace. These by-products contain the highest share of polyphenolic compounds that are known to be associated with beneficial effects on human health. Therefore, they are an attractive source of phenolic compounds for further industrial use in the cosmetic, pharmaceutical and food industries. The use of these phenolics is limited due to difficulties in recovery, high reactivity, complexity of the OMWW matrix and different physiochemical properties of phenolic compounds. This research, focused on OMWW, was performed in two phases. First, different polyphenol extraction methods were compared to obtain the method that yields the highest polyphenol concentration. Twenty-five phenolic compounds and their isomers were determined. Acidifying OMWW, followed by five minutes of ultrasonication, resulted in the highest measured polyphenol content of 27 mg/L. Second, the collection of polyphenolic compounds from OMWW via adsorption on unmodified iron (II, III) oxide particles was investigated. Although low yields were obtained for removed polyphenolic compounds in one removal cycle, the process has a high capability to be repeated.

## 1. Introduction

Polyphenols are naturally occurring compounds found largely in fruits, vegetables, cereals and beverages, and they are characterized by powerful antioxidant activity [1]. They are generally involved in plants as a defence against ultraviolet radiation or aggression by pathogens, parasites and predators [1,2]. In food, polyphenols may contribute to bitterness, astringency, colour, flavour, odour and oxidative stability. Several studies showed that long-term consumption of diets rich in plant polyphenols offered some protection against the development of cancers, cardiovascular diseases, diabetes, osteoporosis and neurodegenerative diseases [3,4]. Bio-based polyphenolic compounds are of increasing scientific interest because of their possible beneficial effects on human health [5,6].

A large source of polyphenols and complex secoiridoids that are not present in other edible plants can be found in the olive industry. Olive oil is the principal fat source of the traditional Mediterranean diet and, due to its high content of polyphenols and monounsaturated fats, has been associated with numerous beneficial human health properties [7]. However, only two percent of the total phenolic content of the milled olive fruit goes into the oil phase, while most is partitioned between the liquid olive mill wastewater (OMWW) (≈53%) and solid pomace (≈45%)―two by-products that olive mills generate using a three-phase decanting system [8]. At the same time, due to this high concentration of organic substances (14–15%) and phenolic compounds (up to 10 g/L) [9], OMWW is known to be one of the most polluting effluents produced by the agrofood industries [10]. Phenolic compounds, especially, exhibit high toxicity towards plants, bacteria, soil and aquatic animals [11]. Due to these negative environmental effects, and because the annual global OMWW production is estimated to be between 10 and 30 million m^3^ [12], different systems have been proposed over the years to treat, minimize or prevent the release of these pollutants [13]. However, the large number of small olive mills across the Mediterranean region make individual on-site treatment options difficult. Moreover, the high phenolic nature of OMWW and its organic content make it highly resistant to biodegradation. Instead of safe OMWW disposal, this matrix (i.e., the components of a sample other than the analyte of interest) can be used as a cheap source of valuable components; thus, it is an interesting opportunity to recover phenols and utilize them as a source in natural food additives, pharmaceuticals or cosmetics.

The recovery of biophenols from OMWW is a difficult task. Phenols are a reactive chemical species, vulnerable to oxidation, conjugation, hydrolysis, polymerization and complexation [9]. OMWW is a complex matrix that offers a reaction medium (water), catalysts (enzymes, organic acids and metals) and substrates (proteins, polysaccharides, metals, small molecular weight reactive compounds and phenols themselves) [14]. Phenolic compounds can bridge or cross-link easily with these compounds [15] or remain attached to cell walls or in the cytoplasmic vacuoles, which all prevent successful extraction. The high variety of phenolic compounds have different structures and different physicochemical properties that makes any attempt to optimize the extraction a difficult task [14]. Several methods to recover polyphenolic compounds were recently investigated with the help of adsorbents [16,17,18], ultrafiltration or nanofiltration membranes [19,20,21,22], microwave assisted solvent extraction [23], drowning-out crystallization-based separation [24] and co-precipitation reactions [25]. A clear review of the different polyphenol recovery methods was prepared by Gullón et al. [26] and Carporasa et al. [27].

Before phenols can be recovered, the quantity and identity of phenols present in OMWW must be determined. Various procedures to determine the phenol content in OMWW have been studied, but most rely on maximizing the recovery of one compound, hydroxytyrosol. Thus, the complexity of the biophenols may be underrepresented [14]. The analysis of phenols in OMWW is similar to methods for phenols from other sources. Solvent extraction is the most common technique to determine the content of phenolic compounds in OMWW, and according to Allouche et al. [28], ethyl acetate is the most effective solvent for the treatment of OMWW under acidic conditions. In this type of extraction, OMWW is pre-treated by filtering solid particles. Fats and oils are removed with n-hexane and then liquid–liquid extraction is performed with ethyl acetate [29,30,31]. Adaptions of this method include (1) adjustment of the OMWW pH to 2 with HCl before extraction [32,33,34], (2) treatment of OMWW with 20% ethanol (*v*/*v*) followed by adjustment of pH to 2 with hydrochloric acid (HCl) before extraction [6] and (3) liquid–liquid extraction with an equal volume of ethyl acetate followed by half volume of hexane [35,36]. Other researchers simply filtered OMWW [37,38,39,40,41,42,43,44]. Delisi et al. [45] and Jebabli et al. [46] compared two methods: (1) ethyl acetate extraction of acidified OMWW, as described before, and (2) lyophilisation OMWW and resuspension in methanol (MeOH). Sedej et al. [47] added MeOH to defatted OMWW, sonicated, centrifuged and evaporated the extract to dryness. The most extensive studies were performed by Jerman Klen et al. [48], who compared five sample preparation methods: filtration, solid-phase (SPE), liquid–liquid (LLE) and ultrasonic (US)-assisted extraction of liquid and solid (freeze-dried) OMWW. The results showed that ultrasonication is a good alternative to conventional solvent extractions, providing higher recoveries at both levels of individual and total phenol yields. In a later study, they also used freeze-dried samples, which were sonicated and extracted with MeOH [49]. Another type of extraction was enzymatic hydrolysis of OMWW using *Aspergillus niger*, *Trichoderma atroviride* and *Trametes trogii*, to release free simple phenolic compounds, combined with ethyl acetate extraction [50]. The phenolic concentration can be measured in different ways. The total phenol content and different phenol classes can be determined by spectrophotometric methods. To determine individual compounds, HPLC, NMR or HPLC-ESI-MS-MS are used [28,29,30,31,32,33,34,35,36,37,38,39,40,41,42,43,44,45,46,47,48,49,50]. Abbatista et al. [51] summarized the methods for the structural characterization of polyphenols in olive by-products.

In this investigation, the polyphenol extraction method that yielded the highest polyphenol recovery from OMWW was studied first. Next, our goal was to collect polyphenolic compounds to valorise OMWW, using them after further clean-up and separation, as a potential polyphenol source for the chemical specialties sector. For this, we investigated the use of unmodified iron (II, III) oxide (Fe_3_O_4_) particles. The key advantage of these iron oxide particles is that they can be easily collected by a magnetic field and therefore deployed into existing technology and infrastructure, providing few barriers to operational uptake [52]. Moreover, they can easily be regenerated and reused, enabling a closed-loop process with several extraction cycles. Conventional techniques such as adsorbing beds are limited because OMWW must run through the whole adsorption bed. This creates a situation that, at the start of the bed, the adsorbent may already be saturated and in equilibrium with the feed, while downstream, the absorbent may not yet be in contact with any solutes [53]. Cleaning saturated adsorption beds is also an intensive process. The use of Fe_3_O_4_ particles also avoids the use of ultrafiltration or nanofiltration membranes, which can be costly to clean or replace after biofouling. The process also avoids mixing of solvents inside the OMWW to collect the polyphenols, in comparison with drowning-out crystallization-based separation microwave assisted solvent extraction.

## 2. Results

### 2.1. Identification of the Polyphenol Content in OMWW from Slovenian Istria

It is known that the composition of OMWW can differ based on the olive types, varietals and provenance. Because of the high variety in polyphenolic compounds, one high-yield extraction technique may be effective for one phenolic compound but not another Therefore, it is not a surprise that several research groups came to different results to determine which is the best extraction technique to obtain the highest polyphenol yield in OMWW. The polyphenolic composition of OMWW obtained from Slovenian Istria was determined via several extraction techniques.

First, different extraction techniques, which were found in literature, were compared to detect the polyphenolic compounds present in OMWW. We used the ethyl acetate and acidified ethyl acetate method since it was claimed to have the best polyphenol extraction yields. We also tested MeOH or MeOH:water (1:1) as an extraction agent. OMWW was also simply filtered. It was shown that simple filtration and lyophilisation with subsequent extraction in MeOH have the same efficiency and give the highest total polyphenol yield [30]. The simple filtration method was also updated by resuspending the obtained residue in MeOH; the final concentration was the sum of the polyphenol concentrations in the OMWW filtrate and MeOH fraction. Quantities of individual polyphenol compounds were measured by LC-MS/MS and expressed semi-quantitatively as counts on the MS detector, whereas the quantification of the total phenol concentration was performed by HPLC-DAD and expressed in mg/mL (Section 4.4). The results are summarized in Table 1.

To allow a quick overview of Table 1, a colour code was applied according to the extracted content of each polyphenolic compound. The lowest concentrations are depicted in dark red, higher concentrations are lighter red, moving towards orange, then yellow and light green, while the highest concentrations are dark green. The lowest extraction efficiency was obtained with the most popular ethyl acetate method (total: 0.95 ± 0.07 mg/mL). Acidifying OMWW before using the ethyl acetate extraction improved the results slightly (1.48 ± 0.10 mg/mL). It is interesting to see that even normal filtered OMWW (total: 3.43 ± 0.24 mg/mL) results in higher extraction yields than the ethyl acetate extract, since simple filtering of OMWW will only lead to the detection of the dissolved polyphenols. With the upgraded filtration, where the residue is dissolved in MeOH, we obtained a total phenol yield of 4.67 ± 0.33 mg/mL. The MeOH extracted weakly bound phenolic compounds such as oleoside isomers, β-oH-verbascoside isomers and caffeoyl-6-secologanoside from the residue. We found that the highest phenol concentrations were obtained with freeze-drying of OMWW and resuspension of the dry matter in MeOH via shaking or ultrasonication (10.1–10.2 ± 0.7 mg/mL). A ten times higher polyphenol content was detected via this method compared to the otherwise popular ethyl acetate method, confirming that the latter is not adequate to determine the polyphenol content in OMWW from Slovenian Istria. Freeze-drying of OMWW and resuspension of the dry matter in MeOH: water (total: 4.99 ± 0.35 mg/mL) was less efficient than resuspension in pure MeOH. MeOH extraction has the highest positive influence on the phenolic compounds oleoside, sacolagonoside, hydroxytyrosol glucoside and 3,4-DHPEA-EDA.

In a second experiment, the influence of ultrasonication and change of pH was tested. OMWW has a pH close to 5 and has a strong buffer capacity; 2M HCl or 2M NaOH was added until the OMWW buffer changed its pH. With HCl, we obtained pH 2; with NaOH, we obtained pH 8. The samples were ultrasonicated for 5, 20 and 40 min to obtain the optimal sonication time. Since no general trend was found between the ultrasonication time and the extracted polyphenol concentration, an ultrasonication time of 40 min was chosen. The results are presented in Table 2.

By only acidifying OMWW, the detected polyphenol concentration increased slightly. In general, acidification had a positive influence on 3,4-DHPEA-EDA and oleuropein aglycone isomers. A more alkali pH decreased the detected polyphenol content. Also here, the alkalization had the most profound effect on 3,4-DHPEA-EDA and oleuropein aglycone isomers, which completely degraded. It is interesting that while most polyphenols degrade, the oleoside and hydroxytyrosol concentration increased. This is probably the result of the cleavage of the oleuropein moieties. The result is in accordance with the phenomenon described by Gentile et al. [54]. In a second step, ultrasonication was applied to the three types of OMWW. Ultrasonication did not seem to have a major effect on OMWW at its natural pH or at pH 8. At acidic pH, however, high polyphenol concentrations were detected (27.6 mg/mL). The total measured phenol concentrations were almost ten times higher than simple filtration of OMWW.

In a last set of experiments, we tested the influence of enzymes on the determined polyphenol content in OMWW. Enzymes are known to cleave bonds within carbohydrates (cellulase, hemicellulase, pectinase) and fats (lipase). Therefore, we used them as a tool to potentially release phenolic compounds, which are bound to such compounds. Different types of enzymes (cellulase, hemicellulase, lipase, pectinase) and their combinations were tested on OMWW. Enzymes were chosen to be compatible with the pH of OMWW. The different treatments showed that enzymatic treatment did not have the expected outcome of releasing high amounts of different polyphenolic compounds in their monomeric form. In general, the amount of detected known polyphenols did not increase and even slightly degraded. Exception was an increase in vanillin (RT 2.4), oleoside (RT 6.5) and caffeic acid (RT 6.7). The main characteristic of the enzymatic treatment was the rise of a large peak within the UV chromatogram (280 nm) at RT 9.04 with *m*/*z* of 242.22 and molecular formula C_11_H_14_O_6_ (see Figure 1). The most straightforward option of the compound identity was an elenolic acid, but the retention time in comparison with other elenolic acid isomers is quite late.

### 2.2. Removal of Polyphenolic Compounds from OMWW by Fe_3_O_4_ Particles

The goal of our research was to valorise OMWW by collecting polyphenolic compounds by adsorption on (un)modified Fe_3_O_4_ particles and desorption in an alcoholic solution. Further processing, clean up or separation can subsequently make OMWW suitable as a new source for polyphenolic compounds in the food, pharmaceutical or cosmetic industries.

The desorbed polyphenol concentrations were measured in MeOH (see Table 3). The first polyphenol extraction with the Fe_3_O_4_ particles yielded what appeared to be a very low quantity of the targeted compounds (0.231 mg per mL of OMWW), especially when compared to extraction in acidified and sonicated OMWW, which yielded over 27 mg/mL (Section 2.1). However, Fe_3_O_4_ particles can be easily regenerated, and reused, enabling a closed-loop process with several extraction cycles. Therefore, we tested a system where these particles were cycled fifteen times between the adsorption (OMWW) and desorption (MeOH) process (each repetition measured separately). The results are summarised in Table 3, where it can be clearly seen that even after fifteen cycles, the Fe_3_O_4_ particles are still taking up polyphenolic compounds, proving their reusability. Most polyphenolic compounds are adsorbed in similar concentrations to the particles even after fifteen cycles. Exceptions are hydroxytyrosol, elenolic acid glucoside and verbascoside, where the desorbed concentrations decrease with each treatment cycle.

To see if the collected polyphenols come from the water-soluble polyphenol fraction or get detached during treatment from other organic matter such as pectin, sugars, fats, proteins or cell walls, the soluble polyphenol content was determined before and after the 15 treatments. Compounds in OMWW such as hydroxytyrosol glucoside and elenolic acid glucoside seem to be collected primarily from particulate organic matter and not the soluble fraction because they have been collected by Fe_3_O_4_ particles in comprehensive amounts, while the elenolic acid and hydroxyltyrosol glucoside content in the water-soluble OMWW fraction did not decrease. The concentrations of caffeic acid and oleuropein aglycone decrease. Apigenin is a compound that is only attached to the organic matter in OMWW, as we do not detect it in the soluble OMWW fraction, but it is extracted by the Fe_3_O_4_ particles in considerable quantity. The content of p-HPEA-EDA and oleuropein aglycone in the OMWW decreases over time in accordance with the attached polyphenolic quantity on the Fe_3_O_4_ particles. The hydroxytyrosol concentrations in OMWW drop much faster than the concentrations that are adsorbed-desorbed by particles, indicating that during the treatment this compound also degrades. Verbascoside seems to turn into β-OH-verbascoside in OMWW during the treatment.

## 3. Discussion

The higher detected polyphenol content, when using MeOH as an extraction solvent instead of ethyl acetate, matched our expectations and was in accordance with the results from Jerman Klen and Mozetič Vodpivec [48], which also indicated the insufficient character of the popular liquid–liquid extraction method. Although the literature evidenced that there is no generally acceptable best solvent for the extraction of polyphenols, it is generally believed that solvents of higher polarity often perform best in terms of polyphenols extraction because of the high solubility of polyphenols in such solvents [55]. An explanation as to why our freeze-drying with resuspension of the OMWW residue in MeOH performed much better than in Jerman Klen and Mozetič Vodpivec [48] could be because we did not acidify the OMWW prior to storage. It is known that pectins can hydrolyse and precipitate under acidic conditions in polar solvents [56]. We suspect that hydrolysed pectins interacted strongly with polyphenols. The fact that acidification with ultrasound extraction was so successful is probably related to a combination of factors. The energy of ultrasonication is known to break bonds, which is why the technique is often used in other matrices to extract different types of compounds. The low pH has a strong effect on fatty acids, protonating their polar head and removing ionic interactions. Alkali pH will have the opposite effect, ionizing all fatty acid groups and making stronger interactions. The phenomenon of phenolic compound degradation under alkali conditions was in accordance with the observations of Friedman and Jürgens [57]. Enzymatic treatment with cellulases, hemicellulases and lipases to break molecular bonds was not as efficient as hoped for. Drawbacks are the long sample preparation times and heating the OMWW to prepare conditions as aligned with enzyme activity as optimally as possible, which also makes it prone to matrix and compound changes.

Removing polyphenolic compounds from OMWW via Fe_3_O_4_ particles is a technique with potential when a multi-step approach is used, by repeating several cycles of adsorption of polyphenols onto the particles and desorbing them into a solvent. This technique will be economically profitable if the Fe_3_O_4_ particles can start a new cycle after desorption and the solvent can be reused by evaporation, leading to a concentration of the polyphenolic compounds in small solvent volumes.

## 4. Materials and Methods

### 4.1. Materials and Instrumentation

Extraction solvents: methanol (MeOH) (Honeywell, HPLC grade, Charlotte, NC, USA) hexane and ethyl acetate (EtAc) (Honeywell, reagent grade). Reagents to adapt the pH of OMWW: hydrochloric acid (HCl, 37%) and sodium hydroxide (NaOH) (Honeywell, reagent grade). OMWW was filtered with 200 nm polyamid (nylon) syringe filters before LC-MS/MS measurements. Iron (II, III) oxide particles (Fe_3_O_4_, 50–100 nm, Sigma Aldrich, St. Louis, MO, USA) were used to collect polyphenolic compounds from OMWW. Solvents for LC-MS/MS analysis: acetonitrile, MeOH and water (Honeywell, LC-MS chromasolv grade).

Enzymes: Tailorenzyme, Herfev, Denmark. Tail 175: xylanase + hemicellulase, pH 4–5, 50–60 °C; Tail 113: pectinase, arabinase, hemicellulase, cellulase, pH 4–5, 45–55 °C; Tailorfood: Tail 157, CAS: 62213-14-3, pectinase, hemicellulase, pH 4–5, 50–60 °C; Tailorfood: Tail 127, CAS: 9001-62-1, lipase (1,3-specific), pH 6.5, 30–40 °C; Tailorfood: CellulX-1L, CAS: 9012-54-8, cellulase + beta-glucanase, pH 4.5–6, 50–60 °C; TailorWine: Extract-01L, CAS: 62213-14-3, pectinase + hemicellulase, pH 3.5–5.5, 45–55 °C.

The liophilizer (Martin Christ, Alpha 1-4 LSCplus, Osterode am Harz, Germany) freeze-dried OMWW; high-performance liquid chromatography coupled to electrospray ionisation and quadrupole time-of-flight mass spectrometer (HPLC-ESI-QTOF-MS, 6530 Agilent Technologies, Santa Clara, CA, USA) was used to qualify and quantify the present polyphenolic compounds. The HPLC equipment incorporated a Poroshell 120 column (EC-C18; 2.7 µm; 3.0 × 150 mm).

### 4.2. Sample Collection

The samples were collected in the first week of October 2019, at the Franka Marzi olive mill (N 45° 30.6588 E 13° 42.2574, Koper, Slovenian Istria). The samples were collected from a three-phase decanter centrifuge. During the three-phase decanting process, olives, from mixed varieties (“Maurino”, “Leccino”, “Buga” and “Istrska belica”), obtained from different cultivars located in the region, are initially washed, crushed and malaxed (churned). Then water is added to a horizontal centrifuge (40–60 L/100 kg fruits weight), separating pomace from the oily mix consisting of the vegetable water and oil. This results in oil, pomace and wastewater fraction. Immediately after sampling, OMWW samples were stored in a freezer (−18 °C). Since these experiments were performed to ultimately find a new way to collect polyphenolic compounds from OMWW on a large scale, OMWW was not acidified as recommended by Jerman Klen and Mozetič Vodpivec [48] because this would not be economically feasible. Since the different steps in the experimental procedures were performed on different days, differences can be found in OMWW composition between experiments. Comparisons made within one experiment were prepared on the same date with the same OMWW.

### 4.3. Extraction Methods to Determine the Polyphenol Content in OMWW

For the results shown in Table 1:20 mL of OMWW was freeze-dried. The residue was shaken (20 min, 200 rpm) using 20 mL MeOH. To remove particles, the MeOH extract was filtered through a 0.2 pore size filter before measurement.20 mL of OMWW was freeze-dried. The residue was shaken (20 min, 200 rpm) using 20 mL of water:MeOH (1:1) mixture. To remove particles, the MeOH extract was filtered through a 0.2 pore size filter before measurement.20 mL of OMWW was freeze-dried. The residue was sonicated for 15 min using 20 mL MeOH. To remove particles, the MeOH extract was filtered through a 0.2 pore size filter before measurement.20 mL OMWW was defatted with hexane (1:1, *v*/*v*). The 2 layers were separated by centrifugation (4000 rpm, 15min) and the hexane layer was removed. Phenolic compounds in the OMWW were three times extracted using a liquid–liquid extraction method by adding ethyl acetate (1:1, *v*/*v*) to the OMWW. The mixture was shaken for 20 min at 200 rpm. Layers were separated by 10 min of centrifugation at 4000 rpm and the ethyl acetate extracts were collected. Ethyl acetate was removed by vacuum evaporation at 40 °C and the oily residue was dissolved in 10 mL of MeOH before measurement [29].The pH of 20 mL OMWW was adjusted to pH 2 using HCl (2 M). OMWW was defatted with hexane (1:1, *v*/*v*). The 2 layers were separated by centrifugation (4000 rpm, 15min) and the hexane layer was removed. Phenolic compounds in the OMWW were three times extracted using a liquid–liquid extraction method by adding ethyl acetate (1:1, *v*/*v*) to the OMWW. The mixture was shaken for 20 min at 200 rpm. Layers were separated by 10 min of centrifugation at 4000 rpm and the ethyl acetate extracts were collected. Ethyl acetate was removed by vacuum evaporation at 40 °C and the oily residue was dissolved in 10 mL of MeOH before measurement [12].Filtration through paper filters; dissolving the obtained residue in methanol and filtering it through 0.2 µm pore size filters. Sum the polyphenol concentration found in the filtrate and the MeOH fraction.Filtration through 0.2 µm pore size filters

For the results shown in Table 2: The pH of 20 mL of OMWW was adjusted to pH 2 using HCl (2 M), raised to pH 8 using NaOH (2 M) or remained at its original pH (pH 5).The pH of 20 mL of OMWW was adjusted to pH 2 using HCl (2 M), raised to pH 8 using NaOH (2 M) or remained at its original pH (pH 5). OMWW was sonicated for 5, 20 and 40 min.

### 4.4. LC-MS/MS Analysis

HPLC-ESI-Q-TOF-MS: An elution gradient of 100% water: formic acid (99.5:0.5, *v*/*v*) (A) towards 100% acetonitrile: MeOH (1:1, *v*/*v*) was used over a period of 20 min (flow rate: 0.5 mL/min; injection volume: 1 µL). The separated phenolic compounds were firstly monitored using a diode-array detector (DAD) (280 nm) and then MS scans were performed in the *m*/*z* range 40–1000 (capillary voltage, 2.5 kV; gas temperature 250 °C; drying gas 8 L/min; sheath gas temperature 375 °C; sheath gas flow 11 L/min). In those conditions, the instruments were expected to provide experimental data with accuracy within ±3 ppm. All data were processed using Qualitative Workflow B.08.00 and Qualitative Navigator B.080.00 software.

The extracts were screened for the range of phenolic compounds previously reported in *O. europaea* L., and their identification was confirmed, based on accurate mass and fragmentation profile, with literature data and analytical grade standards (hydroxytyrosol, luteolin, verbascoside, apigenin, oleuropein) [58]. Tyrosol cannot be detected by MS because of its high ionization energy; its presence in the extracts was confirmed by comparison with the retention times of the tyrosol standard solution using a DAD. Twenty-five phenolic compounds and their isomers were determined by MS: oleoside, hydroxytyrosol glucoside, hydroxtyrosol, elenolic acid glucoside, sacolagonoside, trans p-coumaric acid-4 glucoside, verbascoside, vanillin, demethyloleuropein, rutin, luteolin-O-glucoside, luteolin rutinoside, nuzhenide, caffeoyl-6-secologanoisde, apigenin glucoside, oleuropein, hydroxyl acetate, 3,4-DHPEA-EDA, oleuropein aglycone, oleuroside, p-HPEA-EDA, listroside and apigenin.

The quantification of the total phenol concentration in samples was performed using calibration graphs prepared using tyrosol by HPLC-DAD. The standard deviation between duplicate samples was about 7%. The calibration plots indicated good correlations between peak areas and commercial standard concentrations. Regression coefficients were higher than 0.990. LOQ was determined as the signal-to-noise ratio of 10:1 and was 8.3 µg/mL. For individual polyphenolic compounds found by MS, only semi-quantification was possible since standards of all compounds are needed for full quantification. 

### 4.5. OMWW Treatment with Fe_3_O_4_ Particles

A total of 5 g/L of Fe_3_O_4_ particles were added to 80 mL of OMWW. The solution was shaken for 15 min (200 rpm). The particles were collected at the side of the beaker with a Neodynium magnet, and the OMWW was decanted. Subsequently, 5 mL of MeOH was added to the Fe_3_O_4_ particles. The MeOH was shaken for 5 min (200 rpm) to desorb the polyphenols from the particles. The particles were collected at the side of the beaker with a Neodynium magnet, and MeOH was decanted. The polyphenol concentration was determined with LC–MS/MS. The modified Fe_3_O_4_ and alcoholic solvent could be reused. In this study, the potential of this concept was tested with unmodified Fe_3_O_4_ particles. The scheme depicting the treatment of OMWW by removing polyphenols with Fe_3_O_4_ particles can be found in Figure 2.

## 5. Conclusions

This article discusses the importance of using an appropriate method to determine polyphenolic compounds of interest in a certain matrix. It was found that liquid–liquid extraction with ethyl acetate, one of the most applied methods in OMWW research, had the lowest performance of all polyphenol determination techniques in OMWW from Slovenian Istria. Lyophilisation of OMWW and resuspension in MeOH resulted in the detection of ten times higher polyphenol concentration, while ultrasonication of acidified OMWW resulted in almost thirty times higher polyphenol concentration. 

With a total polyphenol concentration in OMWW of around 30 mg/mL, less than one percent of the polyphenols is removed by Fe_3_O_4_ particles (0.230 mg/mL) in one run. However, the technique’s adsorption and desorption, with help from magnetic collection of Fe_3_O_4_ particles, lends itself to easy repetition. Further research is needed to test different modifications (citric acid, C18, sodium dodecyl sulphate) of Fe_3_O_4_ particles to increase their adsorption efficiency or selectivity.

## Figures and Tables

**Figure 1 molecules-26-06946-f001:**
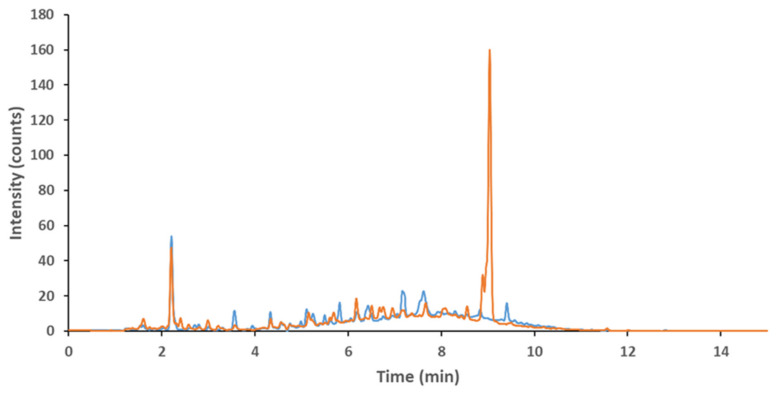
Comparison between OMWW (blue) and enzymatic treated OMWW (orange).

**Figure 2 molecules-26-06946-f002:**
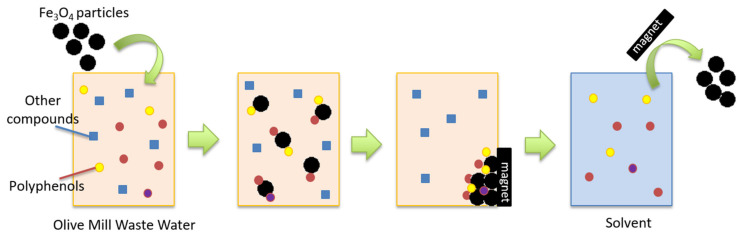
Scheme depicting the removal of polyphenols by the use of Fe_3_O_4_ particles.

**Table 1 molecules-26-06946-t001:** Comparison between different OMWW treatment techniques to determine their polyphenol content. Individual phenolic compounds are semi-quantified (counts on the MS detector), total phenolic concentrations are in mg/mL.

Phenolic Compound	Freeze Dry MeOH, Shake	Freeze Dry MeOH:H2O	Freeze Dry, MeOH, US	Acidfied EtAc	EtAc	OMWW Filtered + Residue	OMWW Filtered
Oleoside isomers	1,445,574	1,110,863	2,289,053	<LOD	<LOD	645,128	328,597
Hydroxytyrosol glucoside	5,66,618	304,694	1,960,374	132,473	209,382	242,807	73,459
Hydroxytyrosol	575,656	604,716	51,584	<LOD	86,154	304,692	247,739
Elenolic acid glucoside isomers	1,029,428	321,183	1,180,572	75,025	65,159	698,553	584,125
Sacolagonoside	3,869,429	132,231	3,324,623	138,370	130,652	445,311	290,300
Trans p-coumaric acid 4-glucoside	34,388	<LOD	<LOD	<LOD	<LOD	70,226	35,977
β-OH-verbascoside isomers	945,209	535,374	998,241	329,091	145,549	528,836	341,598
Vanilin	463,172	<LOD	<LOD	<LOD	<LOD	<LOD	<LOD
Verbascoside isomers	674,206	474,729	633,051	128,349	86,880	295,769	201,862
Demethyloleuropein	344,815	323,718	336,136	22,328	62,148	129,393	94,889
Rutin	161,112	<LOD	93,293	38,197	61,715	<LOD	<LOD
Luteolin-O-glucoside isomers	<LOD	382,343	<LOD	69,310	<LOD	79,061	<LOD
Luteolin rutinoside	<LOD	<LOD	<LOD	58,880	<LOD	114,975	<LOD
Nuzhenide Isomers	538,060	580,076	882,461	78,569	<LOD	359,262	298,155
Caffeoyl-6-secologanoside	610,945	646,526	579,465	61,018	<LOD	343,308	222,196
Oleuropein isomers	453,752	845,967	483,535	143,477	<LOD	745,436	707,012
Hydroxytyrosol acetate	<LOD	<LOD	<LOD	29,961	<LOD	<LOD	<LOD
3,4-DHPEA-EDA	1,751,318	634,046	1,520,368	115,301	125,972	510,237	363,454
Oleuropein aglycone Isomers	1,475,716	749,689	1,440,826	625,458	395,710	1,164,560	1,008,817
Oleuropein/Oleuroside	1,241,321	411,952	385,577	328,181	112,339	508,792	441,105
p-HPEA-EDA	<LOD	<LOD	158,551	<LOD	<LOD	69,982	<LOD
Ligstroside	1,041,186	532,238	1,007,165	130,851	112,607	692,672	663,710
Apigenin	362,484	<LOD	<LOD	37,981	31,983	85,286	<LOD
**Total (mg/mL)**	**10.2 ± 0.7**	**4.99 ± 0.35**	**10.1 ± 0.7**	**1.48 ± 0.10**	**0.95 ± 0.07**	**4.67 ± 0.33**	**3.43 ± 0.24**

**Table 2 molecules-26-06946-t002:** Comparison between different OMWW treatment techniques using pH change and ultrasonication (40 min) to determine the polyphenol content. Total concentrations are in mg/mL, individual phenolic compounds are semi-quantified (counts on the MS detector).

Phenolic Compound	OMWW (pH 2)	OMWW (pH 5)	OMWW (pH 8)	OMWW (pH2 + US)	OMWW (pH5 + US)	OMWW (pH8 + US)
Oleoside isomers	309,970	370399	427,345	1,697,180	289,116	466,507
Hydroxytyrosol glucoside	471,096	284219	21,786	5,227,070	388,312	21,006
Hydroxytyrosol	293,461	260768	68,9939	3,141,870	314,009	341,010
Elenolic acid glucoside isomers	568,689	715803	506,267	2,481,760	717,451	535,038
Trans p-coumaric acid 4-glucoside	99,950	169716	74,408	728,910	114,826	62,571
β-OH-verbascoside isomers	399,555	351,584	223,935	4,395,790	351,633	172,049
Verbascoside isomers	270,546	227,444	186,890	2,518,070	272,648	183,198
Demethyloleuropein	69,378	39,526	37,856	<LOD	63,493	45,920
Nuzhenide Isomers	167,438	138,946	<LOD	379,690	220,018	<LOD
Caffeoyl-6-secologanoside	266,821	268,048	222,335	2,786,090	213,464	214,685
Oleuropein	365,061	355,250	129,169	1,500,900	361,880	162,743
Hydroxytyrosol acetate	<LOD	<LOD	<LOD	177,910	31,639	<LOD
3,4-DHPEA-EDA	780,943	336,025	<LOD	5,456,340	375,536	<LOD
Oleuropein aglycone Isomers	819,442	607,781	<LOD	4,189,430	674,198	<LOD
p-HPEA-EDA	17,413	68,905	<LOD	578,420	70,888	<LOD
**Total (mg/mL)**	**3.83 ± 0.27**	**3.29 ± 0.23**	**1.98 ± 0.14**	**27.6 ± 1.9**	**3.48 ± 0.24**	**1.96 ± 0.14**

**Table 3 molecules-26-06946-t003:** Fifteen subsequent treatments of OMWW with unmodified Fe_3_O_4_ particles. The particles were thereafter desorbed in MeOH. Total concentrations are quantified in mg per mL of OMWW, individual compounds are semi-quantified (counts on the MS detector).

Phenolic Compounds	Polyphenol Content in 1st MeOH Fraction	Polyphenol Content in 15th MeOH Fraction	Soluble Polyphenol Content in OMWW -Before Treatment	Soluble Polyphenol Concentration in OMWW—After Treatment
Oleoside isomers	17,173	18,190	322,365	322,726
Hydroxytyrosol glucoside	3995	6876	71,549	72,656
Hydroxytyrosol	6955	1527	177,540	45,708
Caffeic acid	6246	7554	151,334	93,235
Elenolic acid glucoside isomers	7967	1889	51,519	48,253
β-OH-verbascoside isomers	8275	7939	129,286	179,532
Demethyloleuropein	562	<LOD	24,056	<LOD
Rutin	607	657	10,704	5970
Verbascoside isomers	6474	<LOD	148,867	<LOD
Luteolin rutinoside	1060	1347	17,369	11,588
Caffeoyl-6-secologanoside	7943	7883	128,738	104,644
Luteolin-O-glucoside isomers	4042	3529	23,156	15,171
Oleuropein isomers	<LOD	426	<LOD	<LOD
3,4-DHPEA-EDA	<LOD	423	<LOD	<LOD
Oleuropein/Oleuroside	951	< LOD	49,132	< LOD
p-HPEA-EDA	261	<LOD	7963	3539
Apigenin	1990	1325	<LOD	<LOD
Oleuropein aglycone isomers	369	<LOD	6813	<LOD
**Total (mg/mL)**	**0.23 ± 0.02**	**0.19 ± 0.02**	**3.86 ± 0.12**	**2.68 ± 0.09**

## Data Availability

Data is contained within the article.

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
