# Peer review of "The Valorisation of Olive Mill Wastewater from Slovenian Istria by Fe3O4 Particles to Recover Polyphenolic Compounds for the Chemical Specialties Sector"

_molecules, 2021, doi:10.3390/molecules26226946_

Round 1

Reviewer 1 Report

The presented manuscript described a detailed procedure to valorize the olive mill wastewater (OMWW), which contains the highest content of polyphenolic compounds, known for their beneficial effect on human health. Authors presented comparison of different extraction techniques of phenolics, as well as their novel approach with utilization of Fe3O4 particles for recovery of such moieties. The general soundness of the manuscripts seems to be quite interesting. However, through the paper, some elements lack consistency. The following is some suggestions which would help to improve the paper further.

  1. The introduction part is based mainly on the outdated sources; thus, it does not discuss newer techniques of phenolics recovery (almost 10 references, seen in the first couple of paragraphs are 8 or more years old). I understand that principal background is needed to be discussed, however there has been a lot of development done in the field of polyphenols extraction. I suggest mentioning about variety of physical factors assisted extractions, such as: not only ultrasound assisted, but also microwave assisted extraction; pulsed electric fields pre-treatment. Other techniques based on variety of solvents, influence of time and temperature, ultrafiltration and others. It is crucial to confront other techniques in order to emphasize the novelty and attractiveness of using ferrous particles.
  2. Lines 45-46: there are many studies already proving the beneficial effects of polyphenols on human health.
  3. I realize that acidic environment is the best for phenolics extraction, however how would authors comment the utilization of mild alkaline conditions, which have been suggested in many studies, to hydrolyze and recover bound phenolic compounds? In the introduction section, it is not clearly stated.
  4. Why did authors use methanol in the recovery procedures, since it is considered as toxic solvent? Especially for compounds considered to be used in food/pharmaceutical industry in the future? Besides, ethanol (or bioethanol, according to the circular economy and green chemistry on which grate emphasis have been recently put) gives similar, or according to some studies, pretty high yield of recovery.
  5. The language style should be improved in the lines 157-159.
  6. In line 175 authors mentioned that they have studied the optimal sonification time of the sample treatment. What were the actual levels of this independent variable (range 5-40 min)? In the results section, as well as Table 1, I was not able to find the information which sonication time is presented. What about the results of all sonication times studied? In my opinion, this section lacks the information about obtaining the optimal sonification parameters.
  7. What was the standard deviation of the results in the Table 1? Moreover, what was the number of replications? Both Tables 1,2&3 lacks the +/- S.D. values for the total concentration of polyphenols.
  8. Line 187, please replace the word alkylation with alkalization.
  9. In the lines 196-206 please improve the style of the paragraph. Some of the conclusions/descriptions are misleading.
  10. Lines 304-305: Can authors provide more information on the olives source used in the mill? Harvest period, localization of the trees, etc.
  11. Please improve the section 4.3. – the procedure is not clear enough, and does not match exactly the results section (inconsistency with the Table 1).
  12. Authors frequently used the phrase “matrix”. For readers outside the field, it might be bit unclear without explanation. I would recommend to use a phrase “plant matrix”.

Final remark: I found the idea of using the ferrous particles in adsorption/desorption processes very interesting and feasible. Thus, I am looking forward for further modifications and extended studies. That is why I recommend to implement some scheme/graphical representation of the proposed process. It will for sure show the future potential of the whole concept and will nicely summarize your conclusion section.

Reviewer 2 Report

Dear authors,

I read carefully your manuscript with an increasing interest and actual subject and I like it. I congratulate you for your hard and intense work of testing different procedures of polyphenols separation and analysis from OMWW (real industrial application and working on real industrial wastewater) implying specific reagents for certain extraction methods, advanced analysis apparatuses and a lot of experinces in association with serious and performant analysis competences. It seems that you succeeded to correlate, interpret and present in a clear and logical manner a lot of obtained experimental data and correlate its with already reported findings.

Because it is possible to improve overall presentation and quality of your manuscript, I mention to you few recommendations which will not modify the value and high quality of your findings.

Specific recommendations and comments:

  1. One or two references can be added after the first two sentences of the first paragraph, and also at page 2 line 61 after '...these pollutants .'
  2. Maybe is better to use the adverb form ('firstly') for the term 'first', e.g. line 106 or line 347, or line 203 for 'slightly degraded'.
  3. Line 133, page 3, maybe it is better to use instead of (50:50) the variant of '(1:1)' and also at line 336, page 9, '...water : methanol (1:1) mixture...'.
  4. Line 346, correct the measurement units, i.e. 0.5 mL/min.
  5. Maybe it is better to use the numerical value for 'five' at line 375 and line 377, meaning '5 mL of ....' and so on.

These are a few of my recommendations. Your merit is great due to the complexity and hard work with a lot of necessary analyses, calibration curves to perform and data summarizing in association with not easy analysis to perform on advanced apparatuses/equipments which are not easy to be handle and explore.

Manuscript reviewer

Round 2

Reviewer 1 Report

Recommendation: Publish as is; no revisions needed.

Comments:
As all the reviewer's suggestions have been corrected and explained, my opinion is that this article is acceptable for publication. I am really satisfied with the work the authors did, in order to improve the manuscript - it is a really high level scientific research paper.